# Metal-Based Complexes as Pharmaceuticals for Molecular Imaging of the Liver

**DOI:** 10.3390/ph12030137

**Published:** 2019-09-16

**Authors:** Julia Greiser, Wolfgang Weigand, Martin Freesmeyer

**Affiliations:** 1University Hospital Jena, Clinic of Nuclear Medicine, Am Klinikum 1, 07747 Jena, Germany; julia.greiser@med.uni-jena.de; 2Institute for Inorganic and Analytical Chemistry, Friedrich Schiller University, Humboldtstrasse 8, 07743 Jena, Germany

**Keywords:** liver, molecular imaging, metals, MRI, SPECT, PET, radiopharmaceuticals, contrast agents, metal complexes, diagnostics

## Abstract

This article reviews the use of metal complexes as contrast agents (CA) and radiopharmaceuticals for the anatomical and functional imaging of the liver. The main focus was on two established imaging modalities: magnetic resonance imaging (MRI) and nuclear medicine, the latter including scintigraphy and positron emission tomography (PET). The review provides an overview on approved pharmaceuticals like Gd-based CA and ^99m^Tc-based radiometal complexes, and also on novel agents such as ^68^Ga-based PET tracers. Metal complexes are presented by their imaging modality, with subsections focusing on their structure and mode of action. Uptake mechanisms, metabolism, and specificity are presented, in context with advantages and limitations of the diagnostic application and taking into account the respective imaging technique.

## 1. Introduction

Liver imaging is an essential diagnostic tool which, in addition to biopsy and laboratory tests, provides information on the spatial distribution of liver function and allows the detection and differentiation of benign and malignant liver lesions [1]. Next to the kidneys, the liver is the major organ responsible for clearing the blood by metabolizing and excreting substances and toxins. In addition, the liver acts as the largest gland in the human body, producing bile, cholesterol, and vital proteins, and also acting as hormone regulator [2]. The enormous relevance and scope of liver imaging become immediately clear considering that liver cancers, particularly hepatocellular carcinoma (HCC), represent one of the three most frequent causes of cancer-related deaths worldwide [3], and that other chronic liver diseases, such as cirrhosis, fibrosis, and hepatitis, affect approximately 29 million patients in the European Union alone [4]. The prevalence of these disorders emphasizes not only the diagnostic relevance of liver imaging, but also its role for therapy monitoring purposes, e.g., following chemotherapy or transplantation.

In clinical routine, liver imaging is performed using a wide spectrum of tools including ultrasound, fluorescence imaging, and molecular methods such as magnetic resonance imaging (MRI), single photon emission computed tomography (SPECT), and positron emission tomography (PET). Particularly, the latter three methods employ metal complexes such as contrast agents (CA) or radiotracers, visualizing the interaction between the pharmaceutical agent and a biological target within the liver, e.g., hepatocytes, hepatic transporters, or vascular system [5,6]. The liver parenchyma consists mainly of hepatocytes, responsible for producing proteins and bile, and Kupffer cells, i.e., resident macrophages which are part of the reticuloendothelial system (RES) and contribute to immunity; therefore, specific hepatic uptake of a metal complex can be basically achieved by targeting one of these two cell types.

Metals constitute the largest part of the periodic table of elements, most of them existing in several oxidation states and various isotopes of which many are radioactive. The design of ligands for metal coordination is limited only by imagination, providing access to a rich field of metal-based pharmaceuticals designed for any specific diagnostic problem. The aim of this review was to systematically present the most relevant metal complexes used for liver imaging, relating their field of application to structural considerations, liver uptake, metabolism, and imaging characteristics, as well as to highlight recent developments in the respective fields.

## 2. Metal Complexes in Liver Specific MRI

### 2.1. MRI Principle and Relevant Metals

Magnetic resonance imaging is based on the application of a strong magnetic field around the body, causing hydrogen protons present within the tissues to align due to the fact of their intrinsic magnetic moment. Radiofrequency pulses applied during the measurement cause excitation of the protons. Once the pulse is turned off, the protons relax back to their initial energy state, thereby releasing energy and causing a change in the net magnetization, which can be detected and used for imaging purposes. Signal intensity depends on the relaxation time and proton density; hence, MRI provides images that contain anatomical information based on tissue properties. Elements like carbon (^13^C) and phosphor (^31^P) also produce MR signals, but hydrogen remains the element of choice due to the fact of its abundance in the body [7].

Due to the dipolar interactions between unpaired electrons and the hydrogen nucleus, CA based on paramagnetic metals shortens the relaxation times of protons, thereby increasing signal intensity [8,9,10,11]. While transition metal ions like Mn(II) and Fe(III) can also be used for MRI, complexes of lanthanide gadolinium (Gd(III)) are the most established CA due to the high number of unpaired electrons resulting in a large magnetic moment and high relaxation efficiency [12,13]. Common CA like Gd-diethylenetriamine pentaacetic acid (DTPA)) are considered extracellular CA because they distribute through the vascular system and enhance tissue contrast depending on the perfusion characteristics and relaxation behavior of the chemical environment [14].

The main benefits of MRI consist in the provision of high-contrast images of organs and tissues—particularly soft-tissue structures—with a resolution in the sub-millimeter range [5,6,15], and the obvious advantage of no radiation exposure. However, patients suffering from claustrophobia and—due to the presence of a strong magnetic field—patients bearing metal implants (e.g., pacemakers) often cannot undergo an MRI examination. Furthermore, renal insufficiency is a common contraindication for the use of CA [16].

### 2.2. Gd-Based Liver Specific CA Gd-EOB-DTPA and Gd-BOPTA

Although liver MRI can be performed using unspecific extracellular CA or even without any CA [11,14], contrast-enhanced MRI remains the cornerstone for liver lesion characterization, with liver-specific CA additionally providing relevant functional information [9,11,17]. Since extracellular CA do not enter the hepatocytes, the information gained is limited to a short imaging window in the first minutes, based on different vascularity of tumoral lesion and normal liver tissue [18]. Following bolus injection of an extracellular CA, its distribution is higher in tumoral tissues due to the fact of their over-proportional arterial blood supply [14,19]. In contrast, following bolus injection of a liver-specific CA, the imaging quality is based on two different aspects: the early arterial and venous phase provides information on the lesion vascularity (similar to the information provided by extracellular CA), while the so-called late phase is characterized by hepatocellular-specific uptake of the CA and its subsequent hepatobiliary excretion (Figure 1) [9,14,17,18]. Thus, information on hepatocyte presence and liver function is gained in parallel due to the contrast between the healthy hepatocytes (which take up the CA) and the tumoral tissue, which does not take up the CA [17]. An additional benefit of using liver-specific CA is the possibility of imaging the gallbladder and the biliary tree [11,14]. In general, a major advantage is that images can be acquired in three-dimensional datasets with high temporal and spatial resolution [9,11].

Thus far, Gd-EOB-DTPA (gadoxetate) is the gold standard for liver-specific MRI. Approximately 50% of the administered dose is taken up by hepatocytes and subsequently excreted into the bile via multidrug resistance protein (MRP) 2 [1,20], while the remainder is excreted renally [11,21]. In normally functioning hepatocytes, the uptake is mediated through the organic anion transporters (OATP) OATP1B1 and OATP1B3 as well as via Na^+^/taurocholate co-transporting polypeptide (NTCP), while hepatic lesions usually exhibit a reduced CA uptake [1,11,21,22].

Gd-BOPTA (gadobenate-dimeglumine) [14] exhibits less hepatic uptake (5%) than Gd-EOB-DTPA and is therefore inferior in terms of late-phase parenchymal enhancement [9,11]. However, Gd-BOPTA exhibits a higher enhancement of hepatic vascular structures [11] due to the fact of its increased relaxation enhancement in the presence of albumin [10,23,24]. The maximum parenchymal enhancement is delayed for Gd-BOPTA (0.5–2 h post injection (*p.i.*)) compared to Gd-EOB-DTPA (20 min *p.i.*), meaning that imaging with Gd-BOPTA requires two scans in order to obtain both anatomical and functional information, while imaging with Gd-EOB-DTPA gathers both pieces of information in a single acquisition setting [11,18,25]. The agent Gd-BOPTA is mainly used for liver lesion identification and characterization in early and late phase [19,26], but can also be used for cholangiography [27].

Although the hepatic specificity of Gd-EOB-DTPA and Gd-BOPTA is markedly different, the only structural difference lies in a *p*-ethoxybenzyl group in EOB-DTPA in contrast to a benzyloxyethyl group in BOPTA (Figure 2), allowing the assumption that alkoxybenzylic structures address the liver more efficiently than non-substituted aromatic rings. In the blood, in contrast, Gd-EOB-DTPA and Gd-BOPTA exhibit a similar serum protein binding (approximately 10%) due to the presence of their aromatic rings [28].

Excellent reviews have already highlighted the importance of Gd-EOB-DTPA in contrast-enhanced MRI, emphasizing its role in lesion characterization [11,17,29,30,31] and liver-function quantification [25] in different chronic liver diseases such as non-alcoholic fatty liver disease (NAFLD), liver fibrosis, and liver cirrhosis. Major application fields are also the pre-operative and post-operative evaluation of residual liver function, the volumetric assessment prior to resection [1,32], and the assessment of graft liver dysfunction [11,21]. Similar to Gd-BOPTA, application in MR cholangiography is also possible [20,23,33].

Due to the similarities in hepatic uptake and excretion mechanisms, Gd-EOB-DTPA has repeatedly been compared with ^99m^Tc-mebrofenin, an established liver-specific tracer used in nuclear medicine, and studies have found a good correlation between the results of these two imaging methods [1,34]. Due to the markedly higher temporal and spatial resolution and the absence of ionizing radiation, Gd-EOB-DTPA-enhanced MRI has, in fact, a great potential to be established as an imaging-based liver-function test offering an alternative to established hepatobiliary scintigraphy [1,34,35,36,37,38,39].

### 2.3. Other Gd-Based Liver Imaging Agents

Prior to the establishment of Gd-EOB-DTPA, approaches for liver targeting have focused on Gd-complexes of iminodiacetic acid (IDA) derivatives and lipophilic Gd-DTPA amides [40]. The complex Gd-DISIDA showed insufficient stability [40], while Gd-mebrofenin appeared to, in fact, have liver specificity, but showed a prolonged organ accumulation and limited biliary excretion [41]. The Gd-IDA complexes incorporated in a dendrimer carrier system also showed accumulation in the liver, albeit less than Gd-EOB-DTPA [42].

A new ligand, EOB-DO3A, has recently been developed, featuring a structure strongly related to EOB-DTPA but differing for the presence of a heptadentate macrocylic chelator in contrast to acyclic DTPA. The complex Gd-EOB-DO3A shows higher kinetic inertness than Gd-EOB-DTPA, promising less release of Gd(III) in vivo, while the rapid hepatic uptake (up to 50%) and the biliary excretion observed in mice make it strongly similar to Gd-EOB-DTPA [43].

For tetraazacyclodecane derivatives bearing lipophilic groups like phenyl, butyl or benzyl moieties, it has been shown that there is a distinct difference between anionic Gd species (which are taken up by hepatocytes) and neutral Gd complexes (with limited hepatic uptake), emphasizing the importance of OATP-mediated liver uptake for metal complexes [44]. It has also been demonstrated that the lack of lipophilic moieties, especially in charged species, strongly enhances renal excretion [44]. Other lipophilic complexes exhibiting liver uptake and hepatobiliary excretion include anionic Gd-Cy_2_DOTA [45] and neutral Gd-2,5-BPA-DO3A [46,47].

Another approach is based on the complexation of Gd(III) with bile acids (conjugated to DOTA, DTPA or derivatives thereof [48,49]), based on the fact that bile acid analogues are in general strong targets of the liver. Bile acids are cholesterol derivatives, explaining the high liver uptake of a cholesterol-conjugated Gd-DO3A complex [50].

Conjugates of Gd-DTPA-polyneogalactosyl polylysine [51] and Gd-DOTA-arabinogalactan [52] target the asiaglycoprotein receptor (AGPR), which is expressed exclusively on mammalian hepatocytes. Targeting the AGPR is an established approach for liver targeting with ^99m^Tc serum albumin scintigraphy, a method described later in Section 3.3.

Recently, research has focused on the incorporation of Gd chelates into nanoparticles or vesicles, e.g., liposomes. In general, this approach aims to decrease the toxicity of Gd-based CA and reduce the renal clearance, enabling the use in patients with renal impairment [12]. Additionally, the nanoparticle surface can be functionalized with specific targeting moieties. An early attempt to prepare Gd-DTPA-liposomes for liver MRI with a diameter of 200–300 nm has proved unsuccessful due to the risk of embolism [40]. In contrast, liposome-encapsulated Gd-DTPA of smaller size (50–100 nm) has been shown to successfully accumulate in the RES and slowly clear from the body [53], comparable to a liposome-encapsulated Gd-DTPA-stearate [54]. The Gd-labeled albumin-based nanoparticles were also taken up by the Kupffer cells [55,56]. Novel approaches are focused on enhancing the in vivo stability in order to prevent the release of Gd(III), as well as to enhance T_1_ and T_2_ relaxivities, enabling T_1_–T_2_ dual mode [57], and reduce the detection limit, thereby reducing the dose of Gd [56,58]. In contrast, a different set of nanoparticles (63–70 nm) containing Gd(III) coordinated by lipid-bound DTPA has shown liver uptake and biliary excretion, with most agents being cleared from the body after 24 h [12]. Comparably small (5 nm) Gd-based hybrid nanoparticles formed by bovine serum albumin have also been suggested as candidates for liver-specific MRI [59]. In general, the size of the nanoparticle determines the clearance dynamics [59].

### 2.4. Side Effects of Gd-Based CA and Gd Deposition

Due to the high doses (commonly applied doses are listed in Table 1 at the end of Section 3.4.) of Gd-based CA used for one scan and the well-known toxicity of free Gd(III), the stability of the complex and the potential release of Gd(III) from the ligand have always been a topic of concern when estimating the safety and applicability of a CA. According to the label of Gd-EOB-DTPA, common side effects include headache and nausea and less common effects include dizziness, flushing, pain, and itching [60]. The label of Gd-BOPTA states that more than 10% of the subjects enrolled in clinical trials experienced an adverse reaction, including nausea, headache, injection site reaction, and feeling hot [61]. The frequency of adverse events is comparable to that reported for Gd-EOB-DTPA [62]. An evaluation of six clinical phase IV studies reported that a total of 1.7% patients experienced at least one adverse event, the most frequent of which are dyspnea, nausea, and liver and renal disorders, concluding that Gd-EOB-DTPA is generally safe and well tolerated [63].

A general concern of Gd-based CA administration is the occurrence of immediate hypersensitivity reactions. A study based on 141,623 applications showed a total of 112 cases, with a higher incidence for Gd-BOPTA (0.22%) than for Gd-EOB-DTPA (0.12%). Two cases of severe reaction (i.e., life-threatening) were observed in a total of 84,367 patients, one for each CA [64]. Another study including 10,608 contrast-enhanced MRI examinations showed a 0.5% frequency of adverse reactions following Gd-BOPTA administration and 0.2% following Gd-EOB-DTPA administration [65]. Four cases of severe adverse reactions were observed, three of which with Gd-BOPTA; however, the differences did not prove statistically significant [65] Thus, Gd-BOPTA is considered generally safe, including in the pediatric population [66,67].

A topic of concern associated with the use of Gd-based CA is the risk of developing nephrogenic systemic fibrosis (NSF), especially in patients with renal impairment [8,68,69,70,71,72]. Due to the reduced renal excretion rate, the prolonged blood half-life of the agents promotes dissociation of Gd(III) from the chelator and Gd deposition in tissue, which may cause a variety of pharmacological effects related with Gd(III)-toxicity, like RES inhibition and fibroblast proliferation, ultimately causing NSF [73]. Therefore, in this patient population, MRI is usually performed unenhanced or using reduced doses of CA [73]. Notably, however, Gd deposition has also been observed in the brain [74,75,76], skin [77], and skeleton [78] of patients with normal renal function. A systematic review on Gd deposition in brain and potential neurotoxicity has recently been published [74], in addition to other relevant reviews on the toxicity of Gd-based CA [8,73,79]. A general hypothesis is that—due to the similarity in size—free Gd(III) may compete with Ca(II), therefore affecting the function of Ca(II)-binding enzymes and calcium channels [8,10]. Animal experiments have shown that in depositions, Gd is found partly in the chemical form of intact CA, partly as insoluble Gd, and partly bound to macromolecules [80].

The release of Gd(III) in the body is related with low kinetic inertness of several Gd-based CA, especially in the case of “linear” (i.e., open-chained) ligands like DTPA [10,68,80,81]. As a matter of fact, Gd deposition in the brain following repeated administration of CA has recently been associated even with macrocyclic agents like gadobutrol [82] and gadoteridol [83]. In 2017, the European Medicines Agency (EMA) suspended the marketing of three linear Gd-based CA [84]. The use of macrocylic agents is still allowed; however, only at the lowest possible dose and for justified cases in which unenhanced scans provide images of insufficient quality. The linear liver-specific CAs Gd-EOB-DTPA and Gd-BOPTA are considered to be of medium risk [68] and their use can be continued based on the fact that they are taken up mainly by the liver and meet an important diagnostic need [84]. There is an ongoing debate about whether the benefits of Gd-based CA application generally outweigh the risks, and individual decisions must be made based on medical indication and patient condition, especially considering the kidney function [85].

### 2.5. Liver-Specific CA Based on Mn(II) and Other Metals

Prior to the development of Gd-EOB-DTPA, the Mn(II) complex of DPDP [86]—mangafodipir (Figure 2)—was clinically approved in the 1990s as a hepatobiliary CA. Mangafodipir-enhanced MRI improved the identification of HCC [18,87,88] and more generally the detection, classification, and diagnosis of focal liver lesions (Figure 3) [24,89,90,91,92], including primary and metastatic liver tumors [93,94]. Excellent articles have been published comparing liver-specific Gd-EOB-DTPA, Gd-BOBPTA, and Mn-DPDP with detailed descriptions of detection rates, contrast enhancement on various types of lesions, and influence on the respective acquisition techniques [90].

A 15–20% fraction of Mn-DPDP is excreted via the kidneys, 50–60% via the feces [90,95,96,97]. Following *i.v.* injection, Mn-DPDP is metabolized via dephosphorylation and transmetallation with zinc [96,97]. Demetallation is substantial, and release of Mn(II) occurs rapidly [98,99]. While the ligand is excreted via the kidneys [18], free Mn(II) accumulates in the liver, leading to three-fold T_1_ relaxivity compared to Gd(III) [90,99], after which it is excreted into the bile [100]. Liver parenchymal enhancement begins 1 min *p.i.,* with subsequent clearance into the gall bladder after 15 min [101]. Excretion from the liver is relatively slow (6–23 h *p.i.*) [93]. The agent Mn-DPDP also exhibits accumulation in spleen, pancreas, and kidneys [14,93] due to the release of free Mn(II) in the blood [98,99].

In contrast to toxic Gd(III), Mn is an essential trace element in the body and is excreted via natural ways. The dose of Mn-DPDP applied for a routine MRI (5 µmol/kg [18,90]) contains approximately the same amount of Mn as the normal content of the whole human body [90]. Nonetheless, release of Mn(II) from the complex into the blood and possible neurotoxicity of free Mn(II) has been discussed from the very beginning [87,98,99,100,102,103]. Free Mn(II) has been associated with extensive production of reactive oxygen species (ROS) [104,105], reactive nitrogen species (RNS) [106], and also with mitochondrial manganese superoxide dismutase (MnSOD) mimetic activities [107]. Also, possible adverse cardiovascular responses such as negative inotropy and vasodilation were suspected to occur based on the fact that Mn(II)—particularly if released at high extracellular concentrations—may act as a clinically-relevant Ca(II) antagonist [88,108,109]. However, early studies focused on toxicity and cardiovascular effects have concluded that Mn-DPDP is generally safe in normal clinical use [90,110,111].

In a phase III study, 17% of patients experienced mild to moderate adverse events after exposure to Mn-DPDP [95]. The side effects included flushed face, hot feeling on head and ears, reactive increase of blood pressure, and postural hypotension in cases of overdosing [87,93,97,98,99,103,108,112]. Based on this safety profile, injection of Mn-DPDP must proceed per slow-drip infusion rather than per bolus [14,93,98]. Mangafodipir was removed from the market in 2012 due to the fact of poor sales [87]; however, the recent concerns of Gd-related NSF have sparked a renewed interest in Mn-based CA as alternatives to Gd-based CA [100,113].

A promising new agent enabling MR angiography, exhibiting both renal and hepatobiliary clearance, is Mn(PyC3A) [114,115]. Lipophilic derivatives of Mn(PyC3A) also showed hepatocellular accumulation with Mn(PyC3A-3-OBn) being a potential replacement of standard Gd-based liver CA [116,117]. A liver-specific Mn-based CA (benzothiazole aniline conjugated ethylenediamine tetraacetic acid (EDTA)) of higher stability than Mn-DPDP was reported in 2017. This CA exhibits hepatobiliary and renal excretion (similarly to Gd-EOB-DTPA) and enables tumor localization in a xenograft mouse model, with tumor-to-tissue contrast superior to that observed with Mn-DPDP [118]. Trapping of alkylated Mn(II) complexes into liposomes has also shown hepatic enhancement in rats [100], and an Mn(II) chelating EDTA-based dendrimer (excreted both via the kidneys and hepatobiliary system) has been discussed as a possible CA for imaging of the liver and the hepatobiliary tree [119]. A liver-specific Cr(III)-diethyl-HIDA complex as CA was reported as early as in 1990; however, this compound has shown insufficient liver contrast enhancement [120].

Besides hepatocytes, a major component of the liver tissue are Kupffer cells, i.e., resident macrophages responsible for the metabolism of small particles, lipids, and proteins [121]. The Kupffer cells represent only approximately 15% of the liver but as much as 90% of the whole body RES [2], therefore any CA targeting the RES can be basically considered liver-specific. Imaging of the liver with (ultra) superparamagnetic iron oxides (SPIOs or USPIOs), which typically accumulate in the Kupffer cells, enables the identification of lesions that do not contain Kupffer cells [14]. Sensitivity can be enhanced by doping iron oxides with Mn [122]. Dynamic imaging is possible when the particulate CA can be administered as a bolus, which depends on the particle size [14]. Patients often react with backache to SPIO administration, making slow infusion necessary [18]. In general, colloids do not exhibit notable biliary excretion but they persistently accumulate in the RES over days, until they are destroyed [18]. As these particles are based on iron oxide and do not feature coordinative metal bonds, these agents are not further discussed in this review. Another recently proposed compound with relevant accumulation in the Kupffer cells is a Fe(III) melanoidin chelate. This compound has shown a good contrast between HCC and healthy liver tissue in a mouse model, and represents a potentially safer CA than agents based on Mn and Gd [123].

## 3. Metal Complexes in Scintigraphic Liver Imaging

### 3.1. Scintigraphic Imaging and Relevant Metal Nuclides

In scintigraphy, single photons emitted by gamma nuclides are detected via scintillation crystals and translated into electric signals [6]. While one gamma camera can only record a planar image, single photon emission computed tomography (SPECT) enables three-dimensional imaging via tomographic reconstruction using several gamma heads.

Since collimators are needed to reduce scatter photons, SPECT is less sensitive than positron emission tomography (PET) and requires long scan times, hence limiting the temporal resolution necessary for dynamic imaging [15,124]. However, SPECT imaging continues to be improved with new pin-hole techniques enabling spatial resolution in the sub-millimeter range [15,124]. Spatial resolution also strongly depends on the energy of the emitted gamma photons. ^99m^Tc has become the isotope of choice for nuclear medicine applications, due to its favorable gamma energy of 140 keV, short half-life (6 h), feasibility of pharmaceutical synthesis using ^99^Mo/^99m^Tc generators, and wide availability of established kits [125,126]. ^99m^Tc coordination chemistry has been extensively investigated for decades, offering a wide range of specific chelators for a variety of diagnostic purposes, including tumor targeting, organ-specific imaging (including kinetic modelling), and brain imaging [125,127]. Other metal nuclides used in scintigraphy include ^111^In, ^67^Ga, and ^201^Tl; however, ^111^In and ^67^Ga emit gamma photons of higher energy than ^99m^Tc, hence requiring stronger collimation and resulting in lower image resolution.

The general advantage of nuclear medical imaging over MRI is the much higher sensitivity of the detection systems and the absence of a background signal [1]. In MRI, all protons in the body are excited by the radiofrequency pulse, while in nuclear medicine, the signals are selectively detected from the sites with radiopharmaceutical accumulation. This is the reason why high doses of CA are needed in MRI (in the range of 1–10 mmol), whereas the concentrations of radiopharmaceuticals needed for nuclear medicine imaging usually lie in the pico- or nanomolar range [5,6,15], and are therefore devoid of adverse pharmacological effects. This dosage gap highlights the importance of continuous research focused at increasing the availability and improving the safety profile of CA for MRI.

### 3.2. ^99m^Tc-Complexes for Hepatobiliary Scintigraphy

The ^99m^Tc-labeled derivatives of phenylcarbamoylmethyl iminidoacetic acid (IDA), a lidocaine analogue [35], have been developed since the 1970s [128]. Originally intended for the diagnosis of biliary diseases [2,129,130,131,132,133], the application field has broadened from hepatobiliary scintigraphy (also called cholescintigraphy) to assessment of global and regional liver function and functional liver volume [134], allowing for three-dimensional liver volumetry via SPECT/CT [1]. However, due to the relatively low spatial resolution, this approach is not feasible for liver lesion differentiation [34].

The ^99m^Tc-IDA biscomplexes generally exhibit rapid hepatic uptake, with a biological half-life in blood of only 3–5 min [135]. Cholescintigraphy is usually performed via dynamic imaging. Scans are acquired immediately following *i.v.* injection of the CA, with acquisition timeframes of 2–10 s within the first 2 min. In this early phase, the vascularity visualization is similar to that of extracellular CA. Acquisition timeframes are then prolonged first to 15–30 s and then, after the beginning of the parenchymal phase at between 10–20 min *p.i.*, the timeframes are prolonged to 2 min per frame [129]. Under normal circumstances, excretion into the gallbladder starts at about 15 min *p.i.*, followed by excretion into the duodenum at 20 min *p.i.*, and by complete clearance from the liver parenchyma after 60 min (Figure 4) [2,129]. In cases of bile duct obstruction or reduced duodenal activity, late-stage imaging can be additionally performed between 2–3 h and 24 h *p.i.* [129]. Detailed reviews have been published on the application and relevance of ^99m^Tc-mebrofenin in liver function tests [132,133,136,137].

The ^99m^Tc-mebrofenin, (BRIDA, 3-bromo-2,4,6-trimethyl-IDA (Figure 5)) shows a very high hepatic uptake (>98%) [138] and minimal renal excretion [132,139], and is therefore the most widely used clinical standard for cholescintigraphy [1,140]. Another common radiopharmaceutical is ^99m^Tc-etifenin (2,6-diethyl-IDA, EHIDA, (Figure 5)) [141], which exhibits slightly lower hepatic uptake (82%) than ^99m^Tc-mebrofenin but a longer biliary excretion half-life (37 min) than ^99m^Tc-mebrofenin (17 min). The highest parenchymal activity for ^99m^Tc-mebrofenin and ^99m^Tc-etifenin is visible already at 1 min *p.i.* [141,142]. The ^99m^Tc-IDAs bind to serum albumin and are released when they reach the space of Disse in the liver [35,133,138,139,143]. The hepatic uptake proceeds via OATPB1 and OATP1B3 [1,34,35,134,144], similarly to Gd-EOB-DTPA. The ^99m^Tc-IDAs are excreted unmetabolized into the bile [141] by the same mechanism as bromosulfophthalein and bilirubin [1,132,134,135,143], thus explaining why elevated bilirubin levels can strongly inhibit hepatospecific uptake and increase renal excretion of ^99m^Tc-IDAs [135,143,145]. However, the strength of hepatic uptake inhibition largely depends on the structure of the ^99m^Tc-IDA compound [143]. For example, hepatic uptake of ^99m^Tc-mebrofenin has proven resistant to high bilirubin levels [139,143], which is another reason why this is the most popular tracer in hepatobiliary scintigraphy, although it should be considered that, since albumin is the main plasma carrier of ^99m^Tc-mebrofenin, the liver uptake can be hindered in patients with hypoalbuminemia [140]. Prior to the establishment of ^99m^Tc-mebrofenin and ^99m^Tc-etifenin, various IDA derivatives have been developed as potential hepatobiliary CA, including DISIDA (disofenin [146,147,148,149,150]), HIDA (lidofenin) [139], PIPIDA [143], IOTIDA [151], diethylmonoiodo-IDA [152], as well as IDAs conjugated to phthalein and fluorescein moieties [153], to name only a few [139].

The ^99m^Tc complexes of pyridoxylidene amino acids have also been applied for hepatobiliary scintigraphy [135,154,155], and one of these tracers, ^99m^Tc-PMT, is still used in clinical routine in Japan [156,157,158,159]. Due to the fact of its specific hepatic uptake, it is also possible to identify bone metastasis of HCC [160]. Within 30 min, over 90% of the applied dose of ^99m^Tc-PMT is passed through the liver, with the highest uptake appearing at about 8 min *p.i.* [138]. Liver uptake is mediated by OATPB1 and OATP1B3 just as for ^99m^Tc-IDAs and the complex is excreted into the bile unmetabolized [158].

In an attempt to optimize the properties of the ^99m^Tc complexes as hepatobiliary tracers, extensive studies in structure–distribution relationships have focused on the correlation between substitution pattern, lipophilicity, hepatic uptake, and excretion rates [135,139,143]. Increased lipophilicity due to the fact of *para*-substitution, for example, increases albumin binding of ^99m^Tc-IDA in the blood, which again increases hepatic uptake and decreases renal excretion, while *ortho*-substitution tends to increase renal excretion [138,143,161]. While a high level of lipophilicity generally increases hepatic extraction, it also prolongs the hepatic transit time, thereby degrading the quality of biliary imaging [135,139,143]. Apparently, a greater number of small alkyl substituents (like methyl) is beneficial over a single, larger, and more lipophilic substituent (like isopropyl) in reducing the hepatobiliary transit time, especially if the alkyl substituents are in *ortho*-position [139]. Similar observations have been made for ^99m^Tc-pyridoxylidenephenylalanine [155].

More recent variations of established ^99m^Tc-IDAs include carbonyl ligands [161]. Another approach is to exploit the imaging potential of bile acids, since these are natural ligands recognized by the liver [162]. Thus far, attempts have focused on ^99m^Tc and ^111^In complexes of bile acid conjugates, which have shown hepatobiliary uptake and excretion [162,163].

### 3.3. ^99m^Tc Serum Albumin Scintigraphy and Asiaglycoprotein Receptor Imaging

The asiaglycoprotein receptor (AGPR) is exclusively expressed on mammalian hepatocytes. Damaged liver tissue usually exhibits reduced AGPR expression, which enables estimation of functional liver volume [140,164]. Targeting this receptor with ^99m^Tc-DTPA-galactosyl serum albumin (GSA) conjugated with asiaglycoprotein (AGP) is used to determine the regional hepatic function (Figure 6) [1,35,133,140,165]. This approach has been developed from an original formulation containing galactosyl-neoglycoalbumin (NGA) [166] and is available as a kit in Japan. Excellent reviews have been published regarding the role of ^99m^Tc-GSA in quantification of functional hepatic mass [1,35,133,140].

The ^99m^Tc-labeled galactosylated copolymers based on styrene were developed with the aim to decrease blood circulation time and reduce the potential immunogenicity of ^99m^Tc-GSA. These copolymers show high liver uptake (comparable to ^99m^Tc-GSA) but faster blood clearance [168,169], and have proven successful for the assessment and staging of hepatic fibrosis in mouse models [170]. One copolymer was further developed for use as a SPECT/MRI dual modality imaging agent by adding DTPA moieties into the monomers, thus additionally enabling chelation of Gd(III) [171]. Liver-targeting ^99m^Tc labeled polymers with galactosylated chitosan [172] and dextran [173] have also been developed, with the chitosan-based tracers also exhibiting high renal excretion [172]. Unlike the OATP-mediated uptake of ^99m^Tc-IDAs, all ^99m^Tc-labeled AGPR-targeting complexes are taken up by the hepatocytes via receptor-mediated endocytosis [169]; however, they are not excreted in the bile and, therefore, cannot be used for the diagnosis of biliary diseases [140,169].

Conjugates of NGA labeled with ^111^In have shown hepatic accumulation in mice [174]. Hexavalent lactoside modified with DTPA to chelate ^111^In also exhibited nearly exclusive accumulation in rat and mice liver [175,176,177]. ^111^In complexes of DTPA derivatives bearing aromatic moieties also exhibit mainly hepatobiliary excretion, comparably to their Gd(III) analogs [178,179].

The tracer ^67^Ga citrate can be applied for the detection of HCC, due to the fact of its accumulation in tumors expressing the transferrin receptor [180,181,182]. Since citrate is a weak chelator for ^67^Ga, transmetallation to transferrin occurs in vivo [181]. However, ^67^Ga citrate is quite unspecific, also showing high activity in the blood pool and accumulation in inflammatory tissue [183]. Furthermore, ^67^Ga citrate can also be used to identify liver defects in cirrhosis, comparably to ^99m^Tc colloid [184].

### 3.4. Colloid Scintigraphy

Colloid scintigraphy with ^99m^Tc-sulfur colloid, ^99m^Tc-tin colloid, ^99m^Tc-phytate, and ^99m^Tc albumin colloid is one of the earliest liver imaging techniques [2,129]. Since these tracers mainly target the RES, they do not show notable biliary excretion but accumulate persistently in the Kupffer cells, thereby allowing the visualization of heterogeneities in liver uptake and storage defects, and also enabling scintigraphic liver volumetry [2]. Although of more limited importance compared to hepatobiliary tracers, ^99m^Tc-sulfur colloid scintigraphy is still in use, e.g., for the determination of regional liver function with SPECT [185] and identification of diffuse liver diseases like cirrhosis [186]. Enhanced uptake of colloid in RES regions other than the liver, for example bone marrow and spleen, can indicate cirrhosis [187].

^99m^Tc phytate can be used similarly to ^99m^Tc-sulfur colloid [188,189,190]. Although it is a ^99m^Tc complex, following *i.v.* injection ^99m^Tc phytate rapidly binds to calcium in serum, therefore behaving like a nanoparticle [191].

## 4. Metal Complexes in Liver Specific Pet Diagnostics

### 4.1. PET Imaging and Relevant Metal Nuclides

Compared to scintigraphy, which emerged with the first gamma camera in the 1950s [194], PET has been established for clinical routine only since the early 1990s and is therefore the most recent method of molecular imaging presented in this review.

Positrons emitted by PET radionuclides possess only a short half-life in matter. The distance covered by a positron before annihilation (positron range) ultimately determines and limits the spatial resolution of PET. The positron range lies in the order of a few millimeters and is characteristic for each nuclide since it depends on the positron energy. The annihilation of positrons with electrons results in the generation of two photons with a characteristic energy of 511 keV. These annihilation photons are simultaneously emitted into opposite directions and are detected by scintillation crystals, and the short time window between the detection (coincidence) determines the detection event being processed as “true” or “false” [15,195]. Because of the electronic collimation used in PET (compared to the physical collimation used in SPECT), PET is more sensitive than SPECT and exhibits a higher spatial resolution [6,15,124,196]. Scintillation crystals are arranged in a detector ring surrounding the patient, thereby providing three-dimensional imaging with a high temporal resolution. Hence, the main advantages of PET is the possibility to perform dynamic imaging and, in addition, to obtain a quantification of the tracer activity [6].

The most commonly used metal nuclide in PET is ^68^Ga [197,198]. The main reason for its importance is that it can be produced via a portable, reusable generator system from the mother nuclide ^68^Ge, thus circumventing the need of a cyclotron, required for non-metal PET nuclides such as ^18^F and ^11^C [127,199,200,201,202]. The ^68^Ga nuclide has a half-life of 68 min, allowing time for radiopharmaceutical production [195,202]. Similar to ^99m^Tc species in SPECT, ^68^Ga-tracer synthesis is mainly based on coordinative chemistry, enabling efficient and fast binding of the metal to a specific chelator. There is a continuously growing interest in developing ^68^Ga tracers via kit production (comparably to established ^99m^Tc radiopharmaceutical synthesis) allowing quick, easy, and safe production independently of PET centers or cyclotron production sites [198,200]. Other radiometals of growing interest include ^64^Cu, ^44^Sc, and ^89^Zr, although none of them are currently used in clinical routine [126,198].

### 4.2. Advances in the Development of Liver Specific PET Tracers

Compared to SPECT, PET exhibits superior spatial resolution, therefore the imaging of liver lesions is well established using standard non-metal PET tracers like ^18^F-FDG [203,204] and ^11^C-acetate [205]. Since Ga(III) is a biomimetic of Fe(III), injection of ^68^Ga-citrate selectively leads to the formation of ^68^Ga-transferrin in vivo, which enables the imaging of tumors expressing the transferrin receptor [206]. However, ^68^Ga-transferrin also exhibits a high blood pool activity and accumulation in the lungs, thus resulting in high background activity [207].

Attempts to develop a radiometal complex for functional liver PET imaging reaches far back [208], but to the best of our knowledge none of these potential PET radiotracers has gone beyond the preclinical testing stage. The rationale for developing metal tracers that exhibit comparable hepatic uptake and biliary excretion clearly lies in the translation of structures from well-known liver pharmaceuticals for SPECT and MRI CA. A promising attempt to perform liver function imaging with PET is based on an IDA conjugate of bromocresolphthalein, a suitable ligand for ^68^Ga [208]. The tracer shows high liver accumulation (60% injected dose (ID)); however, biliary excretion is comparably slow and does not exceed 15% ID [208]. While non-modified IDA is per se not a suitable ligand for ^68^Ga due to the rapid demetallation (as shown for ^68^Ga-etifenin [209]), the additional phenolic coordination site on the cresolphthalein moiety provides a more suitable coordination sphere for ^68^Ga [208]. An example of an anionic lipophilic complex is ^68^Ga-*t*-butyl-HBED, which has shown rapid liver accumulation and hepatobiliary clearance in primates [210]. A trichatecholamide analogue of enterobactin, ^68^Ga-3,4-DiP-LICAM, has also been discussed as a possible tracer to study hepatobiliary kinetics [211], whereas ^68^Ga-alizarin has been presented as a possible RES imaging agent [212]. Comparable to ^99m^Tc-sulfur-colloid scintigraphy, ^68^Ga-labeled iron hydroxide colloid presents high liver accumulation [213].

With the intent to target the AGPR, the translation of ^99m^Tc-NGA and ^99m^Tc-GSA, respectively, into a ^68^Ga analogue requires a change in the binding motif to achieve a stable coordination of ^68^Ga. Deferoxamine-NGA labeled with ^67^Ga, developed already in 1992, shows 90% hepatic uptake [214]; therefore, a similar biodistribution can be expected from a ^68^Ga-based PET analogue. While DTPA-GSA is a suitable ligand for ^99m^Tc, ^68^Ga-DTPA-GSA has shown specific liver accumulation but poor stability in vivo [192]. The stability could be sufficiently improved by exchanging the chelator DTPA with the well-known ^68^Ga chelator NOTA (Figure 7) [215]. A similar attempt was undertaken with NOTA-conjugated neolactosylated albumin (LSA), which exhibits 64% hepatic uptake when labeled with ^68^Ga [216]. The ^68^Ga complex of NOTA-hexavalent lactoside, which was developed as a kit for ^68^Ga, shows 20% hepatic uptake and primary excretion through the kidneys [217]. However, similarly to serum albumin scintigraphy, imaging of biliary structures cannot be performed with these tracers, as their biliary excretion is limited [216].

Direct translation of structures from liver-specific MRI CA into PET tracers, by preparing ^68^Ga-EOB-DTPA, has proved unsuccessful due to the fact of tracer instability and rapid demetallation in vitro, providing, again, evidence that DTPA complexes of ^68^Ga are of limited applicability [218]. A new class of lipophilic ligands for ^68^Ga include alkoxysalicyl-substituted DAZA (1,4-diazepane-6-amine) [219], an azacycle that has been established over the lst years as a basis for several ^68^Ga chelators [220,221]. The novel ^68^Ga-DAZA species present a hepatic uptake up to 27%, as well as biliary excretion behavior into the duodenum in ovo, comparably to ^99m^Tc-EHIDA (Figure 8) [219].

Natural liver-targeting structures like bile acids and hepatic transporter substrates labeled with ^18^F [222,223] and ^11^C [224,225] often exhibit specific hepatic uptake and efflux. Labeling of chelator-conjugated bile acids with ^64^Cu has been reported in 2015 [226]. Chelator-free ^64^Cu-chloride can be used to identify HCC due to changes in copper metabolism of tumor tissue [227].

Comparably to the preparation of liposomes encapsulating Gd-DTPA, ^67^Ga-deferoxamin trapped in liposomes also accumulates in the RES of mice liver [228]. Furthermore, rat experiments have shown that ^68^Ga-oxine dispersed in lipiodol is also mainly retained in the liver, although there is some leakage of free ^68^Ga from the lipiodol [3].

## 5. Conclusions

### 5.1. Liver-Targeting Metal Complexes—Influence of Ligand and Metal

A broad spectrum of cholescintigraphic radiopharmaceuticals for scintigraphy and SPECT has been developed in the last decades, including radioiodinated substrates (like ^131^I-labeled rose Bengal) and, in addition to ^99m^Tc complexes, some metal complexes based on less conventional radiometals like ^97^Ru [135,229]. Development of liver-specific MRI CA has focused primarily on Gd complexes. Several strategies in the design of the complexes can be employed to specifically target the liver:
-Use of glycoprotein conjugates, e.g., galactosyl albumin (NGA, GSA) or galactosylated copolymers, to achieve active uptake into the hepatocytes via targeting of AGPR;-Generation of small lipophilic, preferably anionic complexes like Gd-EOB-DTPA, ^99m^Tc-IDAs, ^99m^Tc-PMT, or ^68^Ga-BP-IDA [208], or use of derivatives of natural liver targeting substrates like bile acids [223] and bromosulfophthalein [208], to achieve active uptake into the hepatocytes via transporters like OATP;-Active uptake into the Kupffer cells of the RES via phagocytosis, e.g., using nanoparticles like iron hydroxides or crystalline coated silica, colloids (for example ^99m^Tc-sulfur colloid, ^99m^Tc albumin colloid) or liposomes;-Indirect-targeting mechanisms, e.g., Mn(II) uptake into the hepatocytes following in vivo demetallation of Mn-DPDP.

Extensive studies have revealed the dependency between lipophilicity, aromatic substitution pattern, and charge with respect to hepatic uptake, hepatocellular transit time, and biliary excretion rates for Gd-chelates [47] and ^99m^Tc-IDA complexes [135,139,143,161]. As a general rule, the combination of lipophilic moieties (e.g., aromatic systems or steroid structures) and a negative charge (e.g., provided by a carboxylate) promotes the hepatic uptake [47,135], although several neutral complexes are also taken up by the hepatocytes [43,46,219].

Extent and rate of biliary excretion of any complex strongly depend on liver uptake mechanism, metabolism, and efflux mechanism from the hepatocytes, influencing the possibility to diagnose biliary diseases. In general, RES-targeting substances show little biliary excretion, resulting in prolonged liver accumulation. Also, some pharmaceuticals like the glycoprotein conjugates do not exhibit biliary excretion [140,169]. In general, a molecular weight > 5000 Da represents the upper limit for biliary excretion [135]. Once excreted via the bile, reabsorption from the intestinal tract and enterohepatic circulation must also be considered [135]. Renal excretion represents a competitive way to hepatobiliary excretion, particularly in cases of reduced liver function or bile obstruction. However, renal excretion plays a minor role in the application of nanoparticles exceeding the size limit for glomerular filtration [100] or AGPR substrates, which exclusively address liver cells [133]. Hepatobiliary excretion is prominent for molecules with a molecular weight > 300 Da, while smaller molecules are usually excreted renally [135].

In general, the central metal ion influences the overall charge and structure of the substrate, thereby influencing the rate of hepatic uptake. It has been shown that the ^99m^Tc center is essential for the high hepatobiliary excretion of ^99m^Tc-HIDA compared to HIDA [230]. The case of Mn-DPDP, which is used as a liver-specific CA due to the release of Mn(II) from the complex in vivo, emphasizes the relevance of the central metal regarding biodistribution and complex stability. In contrast to Mn-DPDP, in the case of ^68^Ga or ^99m^Tc tracers the integrity of the complex is crucial to maintain a high liver specificity, since ^68^Ga released from the complex behaves similarly to Fe(III) and is primarily bound by transferrin, resulting in unspecific distribution in the blood pool [206,219], while the release of ^99m^Tc is likely to result in accumulation in non-targeted organs, for example the thyroid.

Since the metal ions used for different imaging modalities (i.e., Gd(III) and Mn(II) for MRI, ^99m^Tc for SPECT, and ^68^Ga for PET) markedly differ in terms of coordination character (e.g., preferred donor number and hardness, metal oxidation states, and charge), liver-targeting ligands have to be specifically designed for each metal ion to create stable metal complexes. For example, IDA is a small, tridendate ligand forming biscomplexes with Ga(III) [209] and Gd(III) [41], which have been reported to exhibit low stability, while the ^99m^Tc compounds are sufficiently stable [141]. Therefore, adaptation of the chelator is often necessary, e.g., from DTPA-GSA for ^99m^Tc to NOTA-GSA for ^68^Ga [215].

Specifically, in the development of MRI CA, it must be noted that the structure of the ligand can heavily influence the relaxation behavior of a Gd-based agent, thereby impacting its applicability and the doses required for sufficient contrast enhancement and image quality. A comprehensive review on this subject was published by Caravan and coworkers, including a detailed discussion of the influence of molecular weight, ligand structure, water exchange rates, rigidity, and other characteristics on the relaxivity of CA [10]. The additional coordination of water molecules to Gd(III) and their influence on relaxation properties have also to be considered, particularly for ligands providing less than nine donors, since Gd(III) prefers a nona-coordination [41]. Herein lies a substantial difference compared to radiopharmaceuticals, since in this case the image quality is determined by the nature and energy of the emission from the metal nuclide, and the coordination sphere does not play any additional influence.

While not extensively presented in this review, particulate metal-based agents for MRI (SPIOs) and metal-based colloids used in SPECT or PET (to a lesser extent), represent an important branch of liver-specific pharmaceuticals of clinical relevance. Radiopharmaceuticals based on non-metal nuclides like ^18^F and ^11^C must also be taken into consideration regarding a future role for PET liver imaging.

### 5.2. Medical Application is Determined by Biodistribution and Imaging Technique

Owing to a much higher spatial resolution, MRI is vastly superior to SPECT in the identification of liver lesions. Therefore, in clinical routine MRI is primarily used for anatomical diagnosis while scintigraphy or SPECT are primarily used for functional diagnostics [34]. Nevertheless, there is a growing interest in using CA enhanced MRI for liver function tests [35].

The different pharmacokinetics of the tracers determine their field of application: As ^99m^Tc-IDAs are cleared very rapidly from the blood and subsequently from the liver, dynamic imaging using SPECT is limited since the temporal resolution is not sufficient for the determination of kinetics and tracer clearance [1]. Therefore, planar dynamic imaging is primarily used in ^99m^Tc-mebrofenin hepatobiliary scintigraphy [133], and the preferred dynamic parameters in this case are hepatic uptake rate and extraction fraction [1]. On the other hand, static ^99m^TcGSA is used for the assessment of regional liver function and functional liver volume with static, three-dimensional images using SPECT [35]. ^99m^TcGSA can also be applied for planar dynamic imaging, but due to the limited biliary excretion and slow liver clearance of ^99m^TcGSA, in this case the parameters of choice are blood-clearance rate and hepatic uptake [1,35]. In general, agents with preferential accumulation in the RES allow the assessment of functional liver reserve and lesion identification, however they do not allow the diagnosis of biliary diseases. In MRI, a dual excretion profile of a CA (i.e., biliary and renal) is preferable for patients with compromised hepatobiliary function, because prolonged retention times in the body can be prevented [47]. These and other considerations (e.g., water solubility) are of little importance for SPECT and PET, because the applied doses of radiopharmaceuticals are low, allowing the application of tracers excreted exclusively via the hepatobiliary system.

In summary, each imaging technique presented here has strengths and limitations, and MRI and nuclear imaging methods remain complementary techniques at this stage of development [6]. MRI with liver-specific CA provides anatomical and functional information based on the blood supply, hepatic uptake and excretion of the agents. SPECT, in turn, provides functional information on hepatic uptake, metabolism, and excretion of a tracer, but the anatomical information remains limited due to low spatial resolution. In SPECT and late-phase MRI, imaging of liver tissue changes (for example metastases) depends on the different hepatic uptake of the agent or tracer, due for example to a reduced expression of transporters in tumoral tissue [41]. Although PET is a powerful diagnostic tool and exhibits superior temporal and spatial resolution compared to SPECT, it has not yet emerged as a dedicated technique for functional liver imaging. Due to the high temporal resolution, PET imaging could additionally provide early-phase vascular imaging, similar to MRI. A unique advantage of PET remains the possibility to obtain quantitative information, which is still limited with SPECT.

Future developments will be guided on one hand by regulatory decisions, most notably the marketing withdrawal (or use limitations) of Gd-based CA, and on the other hand by technical implementations such as increased spatial resolution of new-generation SPECTs and broadened production of ^68^Ga kits for PET as a feasible alternative to SPECT.

## Figures and Tables

**Figure 1 pharmaceuticals-12-00137-f001:**
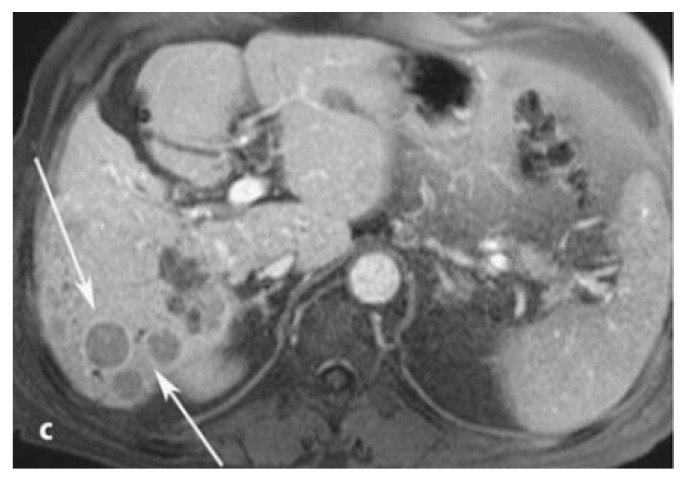
Late phase MRI of the liver obtained after bolus injection of Gd-EOB-DTPA (gadoxetate), showing the presence of hepatocellular carcinoma lesions (arrows). Reprinted with permission from Reference [14].

**Figure 2 pharmaceuticals-12-00137-f002:**
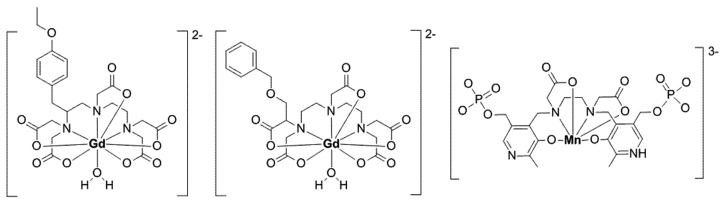
Structure of Gd-EOB-DTPA (left), Gd-BOPTA (middle), and Mn-DPDP (mangafodipir, right).

**Figure 3 pharmaceuticals-12-00137-f003:**
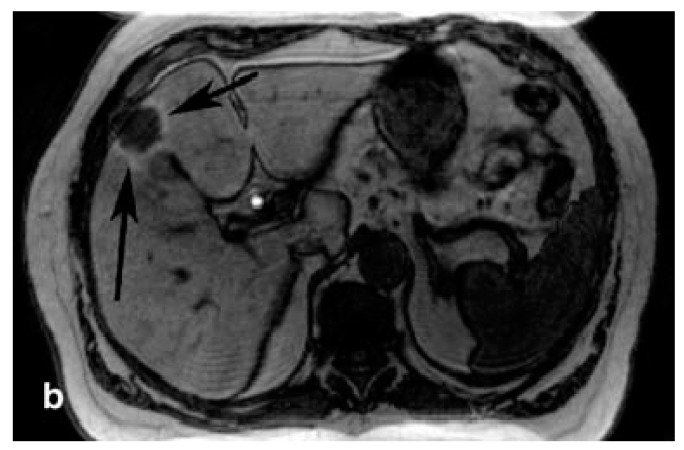
Mangafodipir-enhanced MRI of the liver showing the presence of hepatocellular carcinoma lesions (arrows). Reprinted with permission from Reference [90].

**Figure 4 pharmaceuticals-12-00137-f004:**
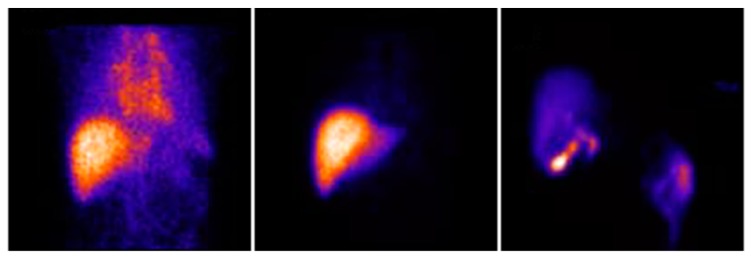
Dynamic planar cholescintigraphy with ^99m^Tc-Mebrofenin showing normal hepatic uptake and blood clearance in the early phase (**left** and **middle**) and biliary excretion in the late phase (**right**). Adapted and reprinted with permission from Reference [132].

**Figure 5 pharmaceuticals-12-00137-f005:**
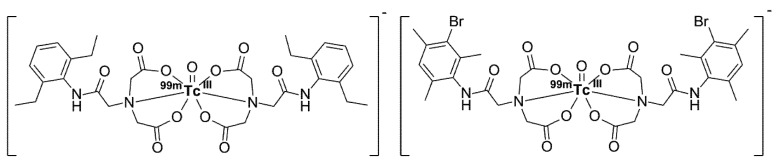
Structure of ^99m^Tc-etifenin (**left**) and ^99m^Tc-mebrofenin (**right**).

**Figure 6 pharmaceuticals-12-00137-f006:**
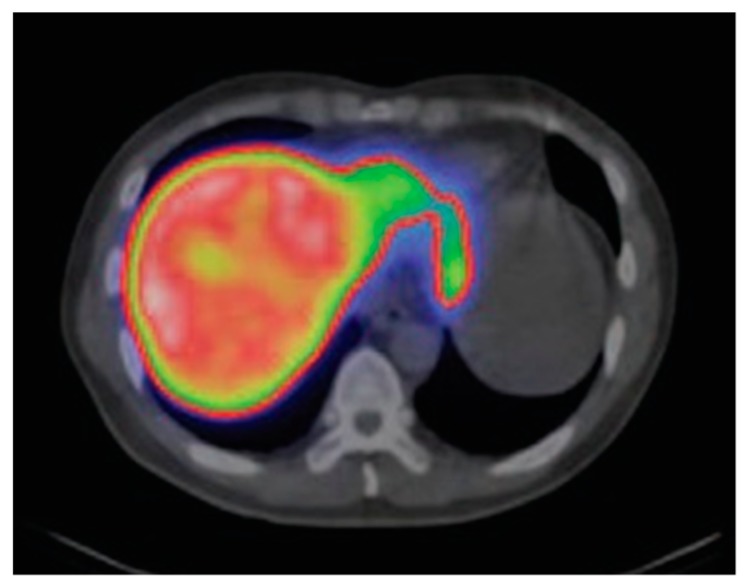
Single photon emission computed tomography (SPECT/CT) scan showing ^99m^Tc-DTPA-galactosyl serum albumin (GSA) accumulation in a fibrotic liver. Reprinted with permission from Reference [167].

**Figure 7 pharmaceuticals-12-00137-f007:**
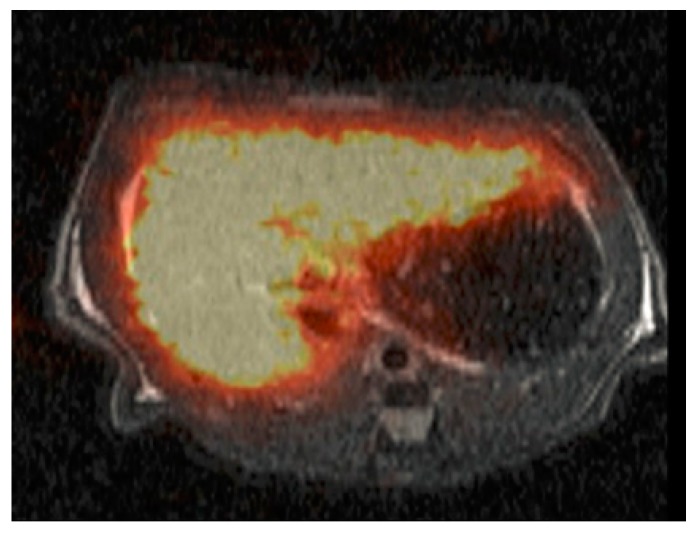
Accumulation of ^68^Ga-NOTA-GSA in rat liver 30 min *p.i.* [215].

**Figure 8 pharmaceuticals-12-00137-f008:**
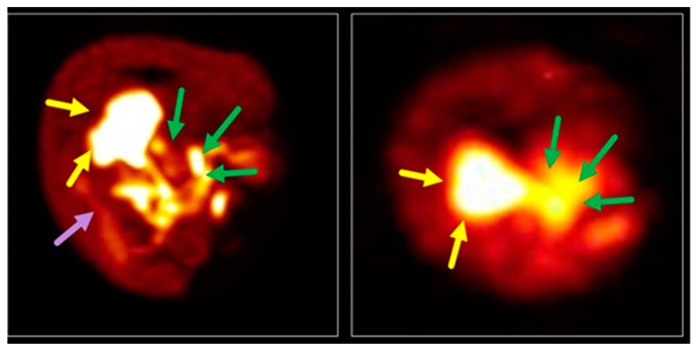
Liver accumulation (yellow arrows) and biliary excretion into the duodenum (green arrows) of ^68^Ga-TEtOHB-DAZA (**left**) compared to ^99m^Tc-EHIDA (**right**) in an in ovo model. ^68^Ga-TEtOHB-DAZA also shows slight accumulation in the kidneys (purple arrow) [219].

**Table 1 pharmaceuticals-12-00137-t001:** List of commercially available metal complexes applicable in MRI and SPECT liver imaging.

	Complex (Commercial Source: Tradename)	Biodistribution ^1^/Target Excretion	Use and Limitations	Applied Dose
**MRI**	Gd-DTPA (Bayer: Magnevist^®^)	extracellular, non-specific, renal excretion	lesion characterization, only early phase visualization	0.1–0.2 mmol/kg [23]
	Gd-EOB-DTPA (Bayer: Primovist^®^, Eovist^®^)	uptake by hepatocytes (50%) with biliary excretion	lesion characterization in early and late phase, cholangiography, Gd-EOB-DTPA can be used for liver volumetry, quantification and liver function test	0.025–0.05 mmol/kg [60]
	Gd-BOPTA (Bracco Diagnostic: MultiHance^®^)	uptake by hepatocytes (5%) with biliary excretion, mostly renal excretion		0.05–0.1 mmol/kg [23]
	Mn-DPDP (GE Healthcare: Teslascan^®^)	dissociation in vivo, uptake of M(II) by hepatocytes (>60%), biliary excretion	lesion characterization, cholangiography, no bolus injection	5 µmol/kg [90]
**SPECT**	^99m^Tc-mebrofenin (Bracco Diagnostic: Choletec^®^, or GE Healthcare: Bridatec^®^)	uptake by hepatocytes (>98%) with biliary excretion	hepatobiliary scintigraphy, liver volumetry, liver function test, diagnosis of chronic liver diseases, no lesion differentiation	0.06 mmol ^2^ per kit [142]
	^99m^Tc-etifenin (ROTOP: EHIDA^®^)	uptake by hepatocytes (82%) with biliary excretion		0.06 mmol ^2^ per kit [141]
	^99m^Tc-PMT (Japan Medi-Physics Co., Chiba [158])	uptake by hepatocytes (>90%) with biliary excretion		6.0 mmol ^2^ per kit [154]
	^99m^Tc-GSA (Nihon Medi-Physics, Tokyo [192])	exclusive uptake by AGPR on hepatocytes, no excretion	liver volumetry, regional hepatic function, diagnosis of chronic liver diseases, no visualization of biliary structures	0.04 µmol ^2^ per kit ^3^
	^99m^Tc-Phytate (Curium: Phytacis^®^)	uptake in Kupffer cells (75% liver), no excretion, phagocytosis		0.03 mmol ^2^ per kit [193]

^1^ Liver uptake can differ considerably from the given values, e.g., in cases of inhibited liver function or high bilirubin levels. ^2^ Since the amount of radioactive ^99m^Tc-complex in all formulations lies in the range of nanomoles, the given number refers to the amount of non-labelled ligand, which is commonly present in large excess compared to the radioactive compound. ^3^ Calculated from the amount of GSA per kit (3 mg) and a molecular weight for GSA of 80.730 Da [192]. DTPA: diethylenetriamine pentaacetic acid, EOB-DTPA: *p*-ethoxybenzyl-DTPA, BOPTA: benzyloxymethyl-DTPA, DPDP: dipyridoxylethylendiamindiacetatediphosphate, mebrofenin: 3-bromo-2,4,6-trimethyl-iminodiacetic acid, etifenin: 2,6-diethyl-iminodiacetic acid, PMT: *N*-pyridoxyl-5-methyltryptophane, GSA: galactosyl serum albumin.

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
