# Peer review of "Metal-Based Complexes as Pharmaceuticals for Molecular Imaging of the Liver"

_pharmaceuticals, 2019, doi:10.3390/ph12030137_

Round 1

Reviewer 1 Report

Molecular imaging of the liver can provide important information for diagnosis, early detection, staging, and treatment of many diseases. This article reviews metal complexes as contrast agents and radiopharmaceuticals for liver imaging. They mainly include some clinically used Gd-, Mn-, Tc-99m, and Ga-68 complexes for MRI, SPECT and PET imaging of the liver. For example, gadolinium-based contrast agent Gd-EOB-DTPA (gadoxetate) can display liver tissue specific properties and is the gold standard for liver-specific MRI. The related structures, imaging characteristics, liver uptake, metabolism, stability, and mechanism have been summarized and displayed.  In addition, some newly developed imaging agents and related structure-activity relations have also been introduced. All the promising results should inspire the readers to further pursue protocol optimization and their applications in liver imaging and related diseases. As summarized in this article, the advantages and limitations of such agents could also guide the research and development of new diverse targeted metal complexes (various metals and ligands) with improved selectivity and specificity for liver imaging. 

Author Response

We thank the reviewer for his or her comments and positive opinion of our article. We believe that the manuscript has been improved even further by the revisions we have done according to the other reviewers’ suggestions, e. g. including table 1 to summarize the most important diagnostic metal complexes.

Reviewer 2 Report

In this review article the authors provided an overview on approved metal-based complexes for molecular imaging of the liver.

The article is well-written and structured.

I would like to suggest to add some tables along the manuscript (i.e. briefly describing characteristics of metal-based pharmaceuticals for MRI, scintigraphy and PET). These tables could be useful for the readers.

Author Response

We thank the reviewer for his or her positive opinion and the very sensible recommendation of adding tables for clarity. Accordingly, we have added Table 1 at the end of chapter 3, summarizing commercial sources, distribution profile, target, use and limitations and applied dose of the most important MRI and SPECT agents.

Reviewer 3 Report

This is a very comprehensive review by Julia Greiser, Wolfgang Weigand, and Martin Freesmeyer. In this review, the authors explain the critical role of metal complexes in medical imaging of the liver. Authors assembled an extensive list of articles on the clinical and preclinical application of metal complexes (contrast agents; CAs) to image the liver function by MRI and nuclear medicine–well implemented and established clinical imaging modalities. The review focuses mainly on the application of metal complexes in medical imaging (Gd-, 99mTc-, and 68Ga-based agents), but metal-free alternatives are also addressed, even though very lightly for simplicity. I recommend publishing this work after a minor revision.

Major comments:

The authors mention to several metal complexes some from commercial sources; it would be easier for the reader if the agents could be listed in a table together with their properties, the commercial source if permitted by the journal guidelines, the role of the agents and limitations.

I recommend showing the chemical structure of most of the agents discussed in this review.

Minor comments:

Authors should check the journal guidelines for using the commercial names of some of the agents as some journals are very strict on the use of commercial nomenclature.
• Line 77-78, “Other common contraindications are renal insufficiency, which may generally prohibit the use of CA, and claustrophobia.” Please rephrase this sentence as it is not clear if the CAs induce renal insufficiency, or if the agents induce associated problems in patients presenting renal insufficiency.
• Line 138-139, "…and also for quantification of the liver functional reserve, thereby offering an alternative to established hepatobiliary scintigraphy.” It is not clear from the text and from the references provided how this was achieved. Are you referring to the use of bimodality MRI/SPECT as a quantitative tool?
• Line 188. How was the number 1-7 mmol calculated, from patients with body weights ranging 50-70Kg? For better comprehension, this number should be compared to other agents routinely used in clinics by PET, SPECT or even CT (iopamidol, for example, is used at even higher doses and it is still considered safe for clinical use). Line 221-224. Gd deposition in the brain with Gadobutrol (macrocyclic EMA approved agents) seems to be related to the recurrent application of the agent in patients (Gianolio et al., 2017, Radiology, 839-849).
• Line 267. It is also important to mention that the free Mn2+ released from MnDPDP, was also associated with an extensive production of reactive oxygen species (ROS), reactive nitrogen species (RNS), and also with mitochondrial manganese superoxide dismutase (MnSOD) mimetic activities. (Ali et al., 1995, Neurodegeneration, 329–334; Patel et al., 1999, Biochim Biophys Acta BBA – Bioenerg, 385–400; Martinez-Finley et al., 2013, Radic Biol Med, 65–75; Liu et al., 2007, Inorg Chem, 8825–8835). In the section about Mn-based contrast agents for MRI, I think it is important for the authors to revise recent work from Gale, Caravan, Tirsco, Platas-Iglesias, et al., on novel Mn-agents for liver imaging. For instance Gale et al., published recently an agent Mn(PyC3A) that is undergoing clinical trials (Gale et al., 2018, Radiology, 865–872; Gale et al., 2015, J. Am. Chem. Soc., 15548-15557).  
